# Regional Agriculture and Social Capital after Massive Natural Disasters: The Case of Miyagi Prefecture after the Great East Japan Earthquake

**Eriko Miyama** 

Faculty of Agriculture, Tokyo University of Agriculture and Technology, 3-5-8 Saiwai-cho,
Fuchu 183-8509, Tokyo, Japan; miyama@go.tuat.ac.jp; Tel.: +81-42-367-5753

**Abstract:** This study aimed to clarify how local agriculture and social capital in disaster-affected areas were transformed by the Great East Japan Earthquake and tsunami in March 2011 and to identify the factors that influenced the transformation of social capital—especially trust—after the disaster. A questionnaire survey was conducted in the Miyagi prefecture's disaster-affected areas. Survey responses were analyzed using descriptive statistics and linear regression analysis with ordinary least squares; the trust index was used for explained variables and personal-attribute disaster-related variables as explanatory variables. The results indicate that regional agriculture was integrated into agricultural corporations or communal management as individual farmers were unable to recover their disaster-related losses. After the disaster, participation in collaborative efforts to manage community resources decreased, while participation in community activities, such as volunteering, increased. Respondents lost trust in the people around them owing to relocation after the disaster and exposure to crime. Steps necessary to maintain or improve social capital in disaster areas include maintaining public safety in the disaster area, securing sources of income, and providing people with interaction opportunities, such as hobby groups. The findings offer practical applications for post-disaster agricultural resource management in developed countries.

**Keywords:** questionnaire survey; social capital; regional agriculture; trust; natural disaster; resilience; Japan



## 1. Introduction

In recent years, natural disasters have become more frequent in many parts of the world, and scientific estimates of their impacts have highlighted the enormous social losses they have caused [1–4]. Furthermore, attention has been focused on the importance of social capital (SC), including the trust and ties among society members, in response to these frequent natural disasters and the recovery process. Aldrich [5] pointed out that SC positively influences resilience; that is, the ability to recover from a disaster. Using the example of an earthquake-affected area in Indonesia, Partelow [6] stated that SC contributes to community recovery by promoting collective action toward obtaining the necessary assistance after a disaster. Iwasaki et al. [7] showed that high relational SC plays a role in relieving disaster victims' anxiety regarding their mental health; an after-effect of a large-scale disaster.

Regarding disaster recovery and SC, studies on the relationships between regional resilience and SC, as well as on individual mental health, have accumulated [8]. However, no study has analyzed the transformation of local agricultural resource management before and after disasters, or the related transformation of SC in developed countries with declining populations. There are three possible reasons why these studies have not been conducted until now. First, obtaining pre-disaster data on SC was difficult. Second, in developed countries, surveys on urban populations and private enterprises have been the norm because only a small percentage of the population works in agriculture and it is not

a major industry. Third, the surveys most commonly conducted in developed countries have focused on urban populations and private companies. However, even in developed countries, agricultural land accounts for a large share of rural areas, and there is a strong need to analyze agriculture from the perspective of land resource management, even in rural regions where the share is small in terms of production value.

Therefore, this study examined how the management of individual farm households was transformed and, more specifically, how trust among other local types of relational SC was transformed after the Great East Japan Earthquake. This event was chosen for analysis because many people were affected by the disaster, in varying degrees of severity and over a wide area. Moreover, the scale of the disaster was such that people were forced to relocate, and forced relocation is a key factor in the transformation of relational SC. Earthquakes and tsunamis are natural disasters that can occur in many parts of the world and are difficult to predict, and their scale of damage has been increasing in recent years. The findings of this study may be useful for managing future reconstruction in disaster-affected areas worldwide.

The first half of this paper addresses the transformation of farm and local agricultural resource management, whereas the second half investigates how people's trust changed from before to after the disaster.

This study aimed to clarify how local agriculture and SC were transformed after a disaster in a large-scale, natural-disaster-affected area and to identify the factors that influenced the transformation of SC, especially trust. Hence, the transformation of SC and the trend of agricultural management recovery in areas affected by large-scale natural disasters were examined via a questionnaire survey conducted in the coastal areas of Miyagi prefecture, the area affected by tsunami during the Great East Japan Earthquake of March 2011.

## 2. Literature Review

Mayer [9] conducted a review on disasters and SC and found that SC positively impacts the disaster recovery process. Mayer [9] stated that SC is a central mechanism for communities to mitigate the effects of disasters and facilitate recovery. Nakagawa and Shaw [10] were among the first to focus on the importance of relational SC in the process of recovery from disasters, conducting comparative studies in Kobe, Japan, and Gujarat, India, to analyze the post-disaster recovery process and identify relational SC as a common denominator to ensure that recovery is sustainable. Aldrich [5] investigated the factors that contributed to the recovery of urban populations after the 1923 Great Kanto Earthquake, the 1995 Great Hanshin Earthquake, and other disasters. The results show that population recovery was faster in areas with higher rates of political participation and where non-profit organizations (NGOs) were established to support recovery efforts. Aldrich also found that in the 2004 Indian Ocean tsunami disaster, villages with both cohesive and consolidated relational SC and individuals with higher levels of SC had access to more support and resources. Although social backgrounds, levels of development, and eras differed, the areas and individuals with well-developed SCs recovered from disasters more quickly and efficiently than those without.

Kawamoto and Kim [11] used a DEA analysis to analyze Japan's post-earthquake waste management. Masud-All-Kamal and Monirul Hassan [12] conducted a qualitative study on the role of SC in the disaster coping and recovery processes of coastal villages in southwest Bangladesh and found that bonding and bridging SCs, in particular, greatly helped villagers during the disaster. Castro-Correa et al. [13] provided details about the roles of institutions and authorities in strengthening, bridging, and linking SC to create trust among community members, as well as among communities, institutions, and authorities, through formal and fluid means, which is necessary to create communication channels. Rayamajhee and Bohara [14] used the post-earthquake disaster recovery process in Nepal to demonstrate how people can mobilize SC to build interpersonal trust and engage in mutually beneficial collective actions. Lee et al. [15] showed through statistics that social

relationship capital-building interventions for older adults contributed to recovery in post-disaster settings. Su [16] found that, in the typhoon disaster that hit Tacloban City in the Philippines in 2013, ties with migrants contributed to household recovery as relational SC.

In the field of medicine, the presence of social relational capital has been shown to reduce the risk of cognitive decline [17]. However, Monteil et al. [18] focused on the negative aspects of SC through a case study analysis of long-term disaster recovery, concluding that in an increasingly diverse society, conjunctive SC may be counterproductive, although bridging and linking SC is important for building social cohesion, which makes an important contribution to sustainable development. They argue that redevelopment measures must be sensitive to the long-term effects of different forms of SC, especially on the building of social cohesion, which is a major contributor to sustainable recovery in a dynamically changing society.

Mayer [9] identified SC as one of the capacities of communities to adapt to uncertain environmental changes, and stated that there are many challenges to operationalizing the mechanism. Discussions on the definition of community resilience and its measurement are important for its practical operationalization. Serfilippi and Ramnath [19] also reviewed and compared community resilience measurement methods and proposed to fill the measurement gap. In recent years, measurement methods have been developed through case studies and theoretical research. Patel et al. [20] reviewed the literature on community resilience and found that the definition is ill-defined. However, rather than seeking a clear definition for the individual components of community resilience, Nguyen and Akerkar [21] organized perspectives for modeling, measuring, and visualizing community resilience from a systematic review of community resilience measurement. Clark-Ginsberg et al. [22] proposed a toolkit of not only validity but also ease of use to measure community resilience. Ostadtaghizadeh et al. [23] conducted a systematic review of tools for measuring community disaster resilience and developed a method for assessing it. They stated that methods for measuring SC should quantify the relative contribution to resilience for each of the social, economic, institutional, physical, and natural domains. Narayan and Cassidy [24] documented both recommended and survey questions to measure SC. The authors also documented the relationship between disaster recovery and SC.

There has also been an accumulation of case study research on the relationship between disaster recovery and SC. Yong et al. [25] showed through empirical research that there are three components of community SC—social trust, interaction with friends, and contact with neighbors—and that these components influence people's attitudes and preparedness behaviors in the event of a disaster. Salim Uddin et al. [26] conducted a participatory study in a village in Bangladesh and showed that community resilience attributes function interactively in determining the foundations and characteristics of community resilience and a clear understanding of network functioning, institutional structures, relationships, and processes driving outcomes. A clear understanding of the processes that drive the results is needed, according to Hudson et al. [27], from a case study on flood risk adaptation in Vietnam, which showed a positive relationship between SC, risk perception, and self-efficacy (self-perceived ability to limit the impact of disasters). Akbar [28], using Yogyakarta Province, Indonesia, as a case study, featured and modeled the impact of partnerships and institutions, education and engagement, and available resources on community resilience. Rivera and Settembrio [29] showed that areas and people with weak SCs are socially vulnerable. Maulana and Wardah [30] stated that maintaining and rebuilding SCs is necessary for community recovery after COVID-19.

Gunderson [31] showed the importance of diversity and cross-scale interactions that contribute to resilience from a socio-ecological systems perspective. Imperiale and Vanclay [32] discussed the need for communities to overcome cultural and political barriers for socially sustainable development through a global culture of well-being and resilience and socially sustainable risk governance. Coles and Buckle [33] argued for linkages with capacity-building programs to enhance community resilience for effective disaster recovery.

Gil-Rivas and Kilmer [34] stated that it is important to focus on community-specific capacity building using an ecological framework to guide the process of disaster recovery.

SC has been shown in much of the literature to be key in sustainable local agricultural resource management. The role of SC in resource management has received much attention since the early 2000s [35,36]. In the 2010s and 2020s, applied case studies on resource management, rural innovation, and the adaptation of climate change response tools in different parts of the world accumulated. Li et al. [37] argued that as societies transition from an agrarian economy to a knowledge economy, a strong SC is one of the necessary ingredients for sustainable rural development and resilience. Barnes et al. [38], through a case study in Papua New Guinea, as a response to the impacts of climate change, found evidence that contact with others in social networks promotes both adaptive and transformative behaviors, consistent with the findings of the model developed and tested by Pakmehr et al. [39] for farmers' behavior in response to water scarcity. This was a structural equation model that explained the adaptive behavior of Iranian farmers. Among other things, they found that collective efficacy enhanced the predictive power of the model. Saptutyningsih et al. [40] also showed, from a case study in Indonesia, that SC, consisting of trust, community involvement, and personal relationships with other villagers, plays an important role in the climate change adaptation process. Musavengane and Kloppers [41], through a case study in South Africa, stated that SC is an effective investment in building community resilience. However, King et al. [42] focused on the negative aspects of SC and suggested that in order to effectively implement rural innovation in communities, a better understanding of SC and the construction of trust is needed.

All these studies were conducted in rural areas of developing countries or urban areas of developed countries, with limited findings related to rural areas of developed countries, where significant population decline and associated sustainable resource management are the primary challenges. In addition, most of these studies examined the role of structural SCs, and few studies have analyzed the transformation of cognitive SCs. In rural areas, the waterways and farm roads necessary to maintain fields often have to be managed jointly by multiple farmers, and the existence and quality of the community are important in their maintenance. When community restructuring occurs after a major disaster due to large-scale displacement or the death of residents, it is essential to examine what changes have occurred in relational SC within the community to manage local agricultural resources in rural areas in developed countries.

Based on the above, this study focuses on the changes in agricultural management that affect local agricultural resource management before and after a disaster, and the transformation of trust, one of the representative cognitive SCs, in the coastal areas of Miyagi prefecture. This area was affected by the Great East Japan Earthquake and is in the process of recovery. It is also facing a population decline.

## 3. Research Methods

### 3.1. Sampling Methods

A preliminary survey was conducted via face-to-face interviews with five people with experience in agriculture and two agricultural corporations in Sendai City, Miyagi prefecture, in April 2015, with the cooperation of the Japan Agricultural Cooperative (JA), Sendai (In Japan, the JAs provide a range of services related to agriculture, including agricultural guidance, financial assistance, insurance, material sales, and provision of sales channels. They are commissioned by the government to conduct surveys, etc., because of their intimate knowledge of local agriculture and farmers). The interviewees were selected by the JA staff according to the agricultural management in the area in terms of scale and items. The prototype of the questionnaire was used to evaluate and improve the questions. Next, the target population for the study was determined. Based on the results of the 2010 Census of Agriculture and Forestry in the coastal area of Miyagi prefecture, designated areas were selected where the percentage of farmers exceeded 10% of the residents (areas with both farm and non-farm households), both at the time of the Great

East Japan Earthquake, in March 2011, and at the time of sending the questionnaire in September 2016. To mail the questionnaires, the Town Plus service provided by the Japanese post office was used. This service is often used to send direct mail, posting mail to all mailboxes in designated areas. The distribution areas were selected from the coastal areas of Shichigahama-cho, Miyagi-gun, Miyagino-ku, Sendai-shi, Wakabayashi-ku, Sendai-shi, Natori-shi, Yamamoto-cho, Watari-gun, Watari-cho, Watari-gun, Miyagi prefecture, and areas adjacent to the coastal areas where the percentage of farmers exceeded 10% (Figure 1).

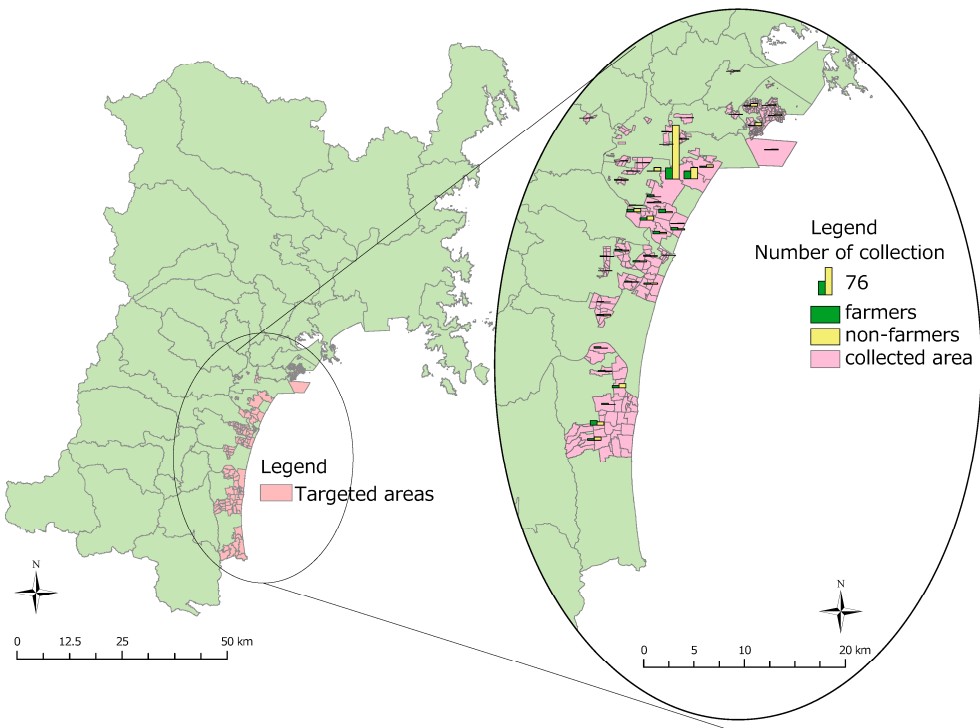

**Figure 1.** Areas where questionnaires were collected.

A total of 5155 questionnaires were distributed, mainly to residents in the coastal areas of Miyagi prefecture (Table 1). The questionnaire comprised the following items: farming status before and after the earthquake and intentions to farm in the future; the extent of damage caused by the earthquake and responses to it; SC status (social contacts and trust in people); and personal attributes. Iwasaki et al. [7] was used as a reference for the questionnaire items on SC. All the respondents were informed at the beginning of the survey about the use of the collected information and the protection of their personal information. The survey was conducted in accordance with the Code of Ethics.

**Table 1.** Summary of questionnaire collection factors; *n* = 5155 surveys.

| Survey Distribution Steps | Distribution Details |
|---|---|
| Distribution method | Sent by mail (Town Plus service by Japan Post) |
| Collection period | 3 September 2016–14 April 2017 (98.2% were collected in September 2016) |
| Number of surveys collected (farmers) | 515 (178) |
| Collection rate | 9.99% |
| Targeted areas | Coastal and surrounding areas where the farmer households comprise more than 10% of all the households in Shichigahama-cho, Miyagino-ku, Wakabayasi-ku, Natori-shi, Yamamoto-cho, and Watari-cho |

Next, the summarized results of the questionnaires by item confirmed the transformation of local agriculture and changes in indicators related to SC and clarified the factors that affect individual trust after a disaster through regression analysis using the least squares method. Based on the analysis results, suggestions for community development in the event of a large-scale disaster that requires the relocation of victims from their residences were discussed.

### 3.2. Data Analysis Methods

A linear regression analysis was conducted using the least squares method, with the trust variable as the explained variable and the variables listed in Table 2 as explanatory variables to confirm the transformation of trust, a representative cognitive capital among relational SC factors. The model used is shown in Equation (1). The explanatory variables consist of control variables based on general personal attributes, such as gender, age, and income; disaster-related variables, including moving history, evacuation history, amount of damage, and whether a person was a victim of a crime since the disaster; and neighborhood-related variables, such as participation in volunteer activities and social gatherings. In Equation (1), $\alpha$ is the intercept, $\beta_k$ represents the regression coefficient of the explanatory variables, and $\varepsilon$ is the error term. The dependent variable was the trust index, which was represented by the total score of the responses to the four questions on trust (Table 3).

**Table 2.** Explanation of variables.

| Variables | Description |
|---|---|
| Trust | Total score of responses to questions in Table 3 in 2016: 0–4. |
| Gender | Gender of respondents: male = 1, female = 0 |
| Farm | Farmer = 1, non-farmer = 0 |
| Community farm | Participating in community farming (in 2010) = 1, others = 0 |
| Marital status | Single-person household = 1, others = 0 |
| Age | Age of the respondent (in years, 2016) 10s = 1, 20s = 2, 30s = 3, 40s = 4, 50s = 5, 60s = 6, 70s = 7, 80s = 8, 90s and above = 9 |
| Income | Household income in 2015: Less than JPY 1 million = 1, JPY 1–2 million = 2, JPY 2–3 million = 3, JPY 3–4 million = 4, JPY 4–6 million = 5, JPY 6–8 million = 6, JPY 8–10 million = 7, JPY 10 million or more = 8 |
| Move | Different residential zip codes immediately before the 2011 earthquake compared to at the time of the survey response in 2016 = 1, same zip code in 2013 and 2016 = 0 |
| Evacuate | Have evacuation experience = 1, no evacuation experience = 0 |
| Volunteer | As of 2016, volunteering = 1, not volunteering = 0 |
| Hobby | As of 2016, participating in a tea party or other hobby = 1, not participating = 0 |
| Damage | Amount of damage to houses and household goods during the year after the earthquake None (JPY 0) = 1, JPY 10,000–500,000 = 2, JPY 500,000–1 million = 3, JPY 1–3 million = 4, JPY 3.1–5 million = 5, JPY 5–10 million = 6, JPY 10,100,000 or more = 7 |
| Crime | Victims of burglary or trespassing immediately after the earthquake, including at their homes in the affected areas while they were away from home = 1; not victimized = 0 |

**Table 3.** Questions about the trust index.

| Question | Points |
|---|---|
| I often leave the door unlocked when going out | yes = 1, no = 0 |
| I often lend money or things to friends | yes = 1, no = 0 |
| I can basically trust my neighbors and acquaintances | yes = 1, no = 0 |
| In general, I think people only act in their own best interests | yes = 0, no = 1 |

Based on the descriptive statistics of the questionnaire results, explanatory variables that may influence trust, a cognitive SC, were selected. In addition to general attributes of individuals, such as gender, age, marital status, and income, variables related to agriculture (farm and community farm), the impact of variables related to disaster (evacuate, move, damage, and crime), and networked SC (volunteer and hobby) on trust were tested. Of these, community farming is a form of cooperative farming by local residents supported by the government in Japan, where the farming population is decreasing. Community farming is included in the explanatory variables because it needs and fosters social relationships.

$$
\begin{aligned}
Trust_i = \alpha + \beta_1 Gender_i + \beta_2 Farm_i + \beta_3 \text{Community\_farm}_i + \beta_4 Single_i + \beta_5 Age_i + \beta_6 Income_i + \beta_7 Move_i \\
+ \beta_8 Evacuate_i + \beta_9 Volunteer_i + \beta_{10} Hobby_i + \beta_{11} Damage_i + \beta_{12} Crime_i + \varepsilon
\end{aligned}
\tag{1}
$$

## 4. Results of Analysis and Discussion

### 4.1. Questionnaire Response Rate

A total of 5155 questionnaires were sent out using the post office's Town Plus in September 2016, and 515 questionnaires (of which 178 were from farm households) were collected by April 2017. The collection rate was 9.99%.

### 4.2. Residents' Situations Regarding the Disaster and Methods of Compensating for Losses

In the coastal areas of Miyagi prefecture, many households suffered damages exceeding JPY 10 million to their houses and household goods (Figure 2). Among farm households, more farmers in the southern part of the prefecture reported a greater loss of agricultural machinery and materials than those in the northern part of the prefecture (Figure 2).

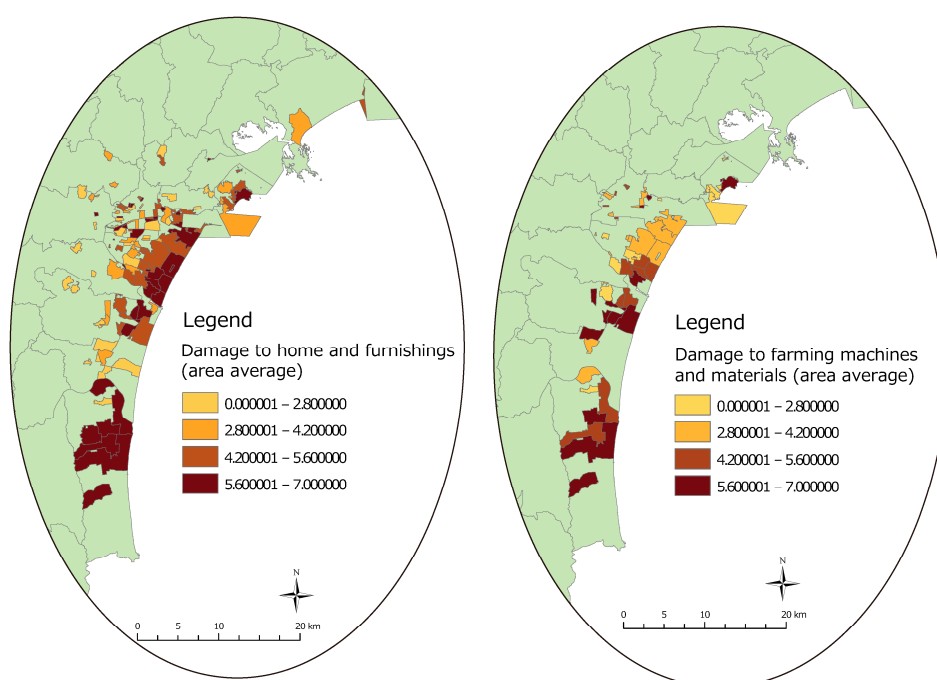

**Figure 2.** Geographic distribution of average damage by area (category). The amount of damage during the first year after the earthquake: (1) none (JPY 0); (2) JPY 10,000 to less than JPY 500,000; (3) JPY 500,000 to JPY 1 million; (4) JPY 1,010,000 to JPY 3 million; (5) JPY 3,010,000 to JPY 5 million; (6) JPY 5,010,000 to JPY 10 million; and (7) JPY 10,001,000 or more. The arithmetic mean of the selected numbers was calculated for each area.

Regarding how the victims compensated for their losses, most households chose to use savings, followed by donations. Substitutes were distributed based on the location of the house or farmland and the type of management. Some households were unable to receive subsidies depending on the type of damage, and respondents from the households

indicated a need for more fairness in the survey's free-text entry. For example, the setting of subsidies made them advantageous for joint farm management and disadvantageous for individual management (according to interviews with individual farmers in Sendai City, April 2015). Additionally, because of the wide range of affected areas, distributing subsidies equally to all households was difficult. Even in adjacent areas, the availability of subsidies differed from one street to the next.

Regarding damage related to agriculture, 84% of the respondents, who were farmers, reported that their farmlands, agriculture-related equipment, and materials were damaged. In the preliminary interviews, farmers reported some cases in which the respondents shifted to communal management because rebuilding lost machinery and equipment in individual management was difficult. The questionnaire survey revealed that the share of individually managed farms decreased significantly after the earthquake, whereas the share of community farms and agricultural production corporations increased (Figure 3). Furthermore, 23% of respondents who are currently farming wanted to leave farming within the next 10 years (Figure 4). The most frequently selected reason for leaving farming after the earthquake was the inability to provide machinery and equipment to replace what had been lost in the disaster. This result suggests that the tendency to share management was stronger in areas that were more severely affected. As for farmland that was no longer being cultivated, the most common reason was asking a corporation to cultivate it (Figure 5).

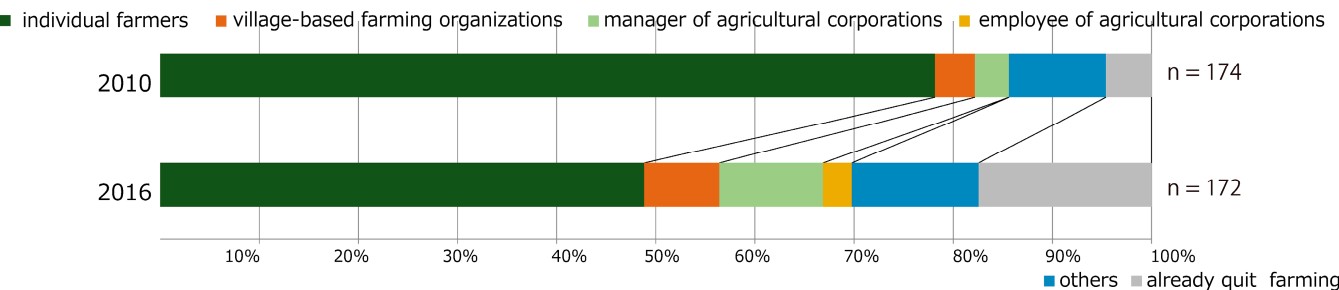

**Figure 3.** Change in agricultural management before and after the earthquake.

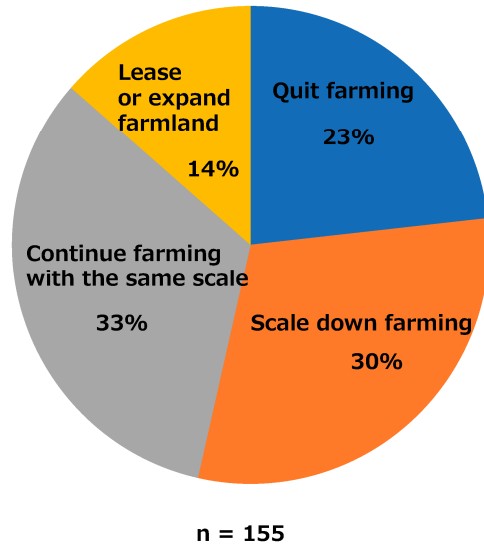

**Figure 4.** Intention to farm in the next 10 years (as of 2016).

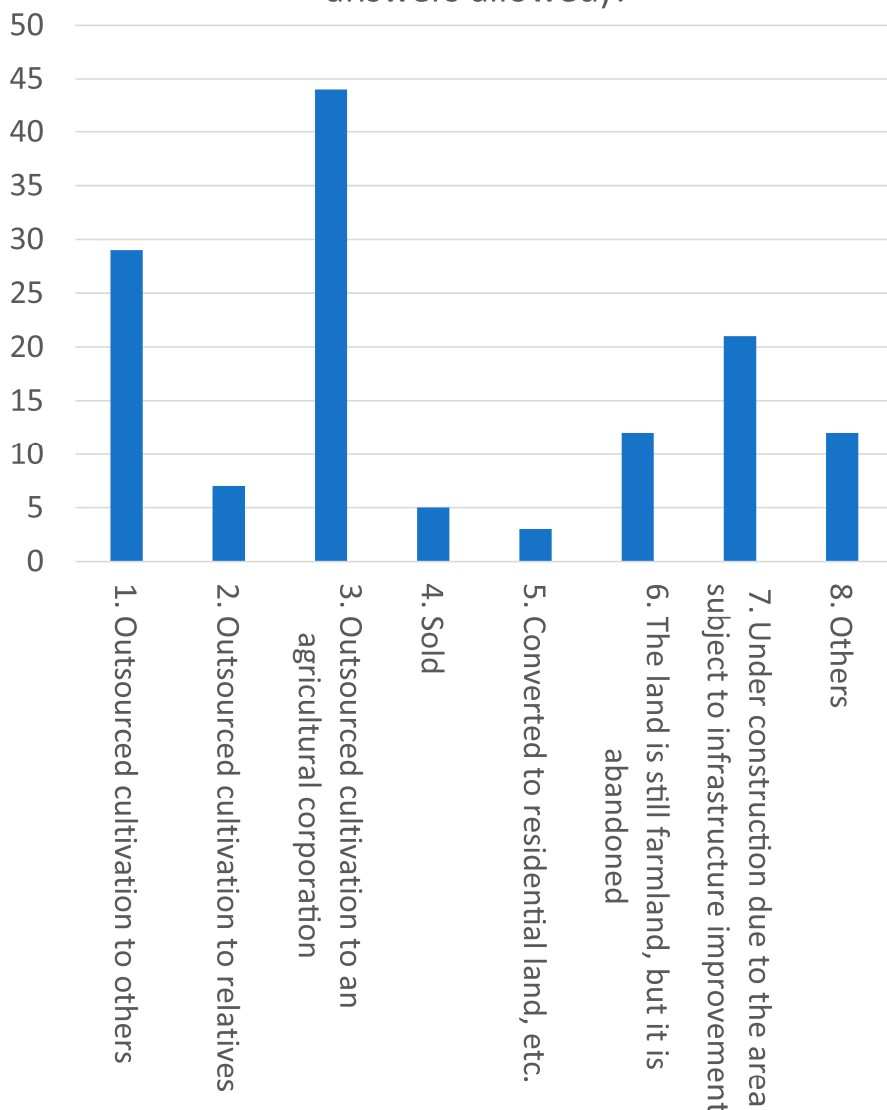

**Figure 5.** Handling of farmland that has stopped being cultivated (*n* = 133).

　　　The results of these analyses indicate the enormous scale of the disaster, such that financial support from the government related to agricultural recovery proved insufficient; subsequently, many farmers left their profession for economic reasons, resulting in the consolidation of farmland into corporations. The pressure for farmers to leave farming and incorporate had been relatively high even before the disaster, as many in the area were originally dual-income farmers. The disaster is thought to have reinforced this trend. The implications of this are both positive and negative for reconstruction. First, from the perspective of agricultural production, production has become more concentrated and efficient. Second, from the perspective of SC in the local community, the population related to agriculture and local resource management has decreased.

### 4.3. Changes in Social Capital

In the question related to relationships with neighbors, the largest number of respondents (47%) answered that they had acquaintances from before the earthquake living in more than 20 households in their neighborhoods. This suggests that many of them returned to their original areas after the evacuation (Figure 6). However, more than 10% of the respondents said that they did not have any acquaintances in the neighborhood of their current residence before the earthquake, suggesting that a certain percentage of neighborhoods had changed dramatically since the earthquake. Therefore, support will be necessary for new communities to be constructed.

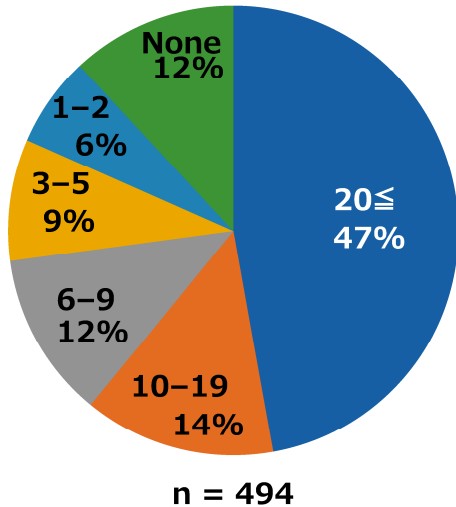

**Figure 6.** Number of acquaintances living in the neighborhood after the earthquake.

The frequency of participation in activities to maintain local agricultural resources, such as mudding waterways and maintaining farm roads, showed a decreasing trend after the earthquake. This finding may reflect an increase in households that have stopped farming (Figure 7).

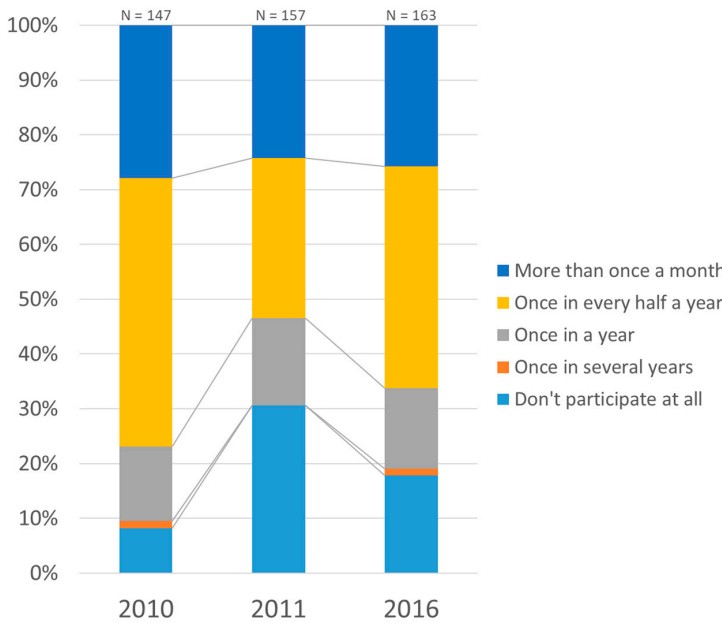

**Figure 7.** Frequency of participation in community outreach activities to maintain local agricultural resources.

Regarding questions related to trust in people, in general, and in local security, there were many areas where the local community became unstable after the earthquake. Immediately after the earthquake, people in the affected areas tended to be more vigilant about their surroundings, as they were often victims of crime (Figures 8 and 9). Particularly in areas where the number of households decreased owing to tsunami damage and/or wherein the number of vacant lots and debris disposal sites increased, people expressed their concerns about children's playgrounds, shopping places, and security at night in the free description column.

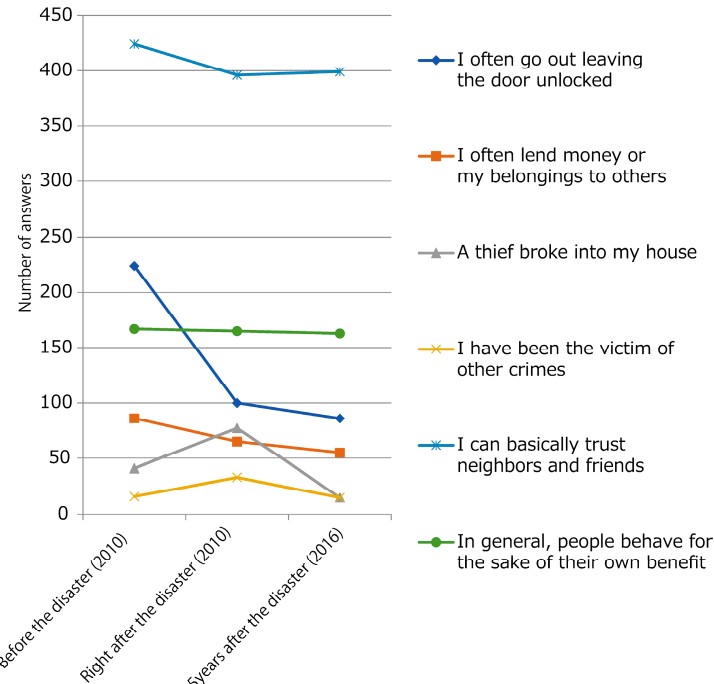

**Figure 8.** Changes in the number of responses related to trust.

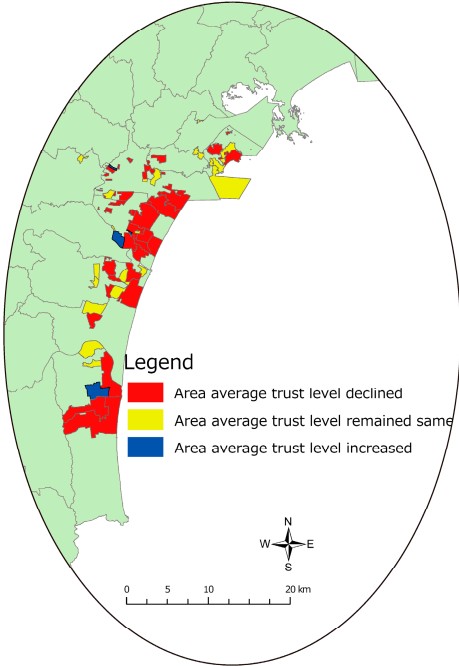

**Figure 9.** Regional distribution of changes in trust indicators.

Regarding participation in social activities before and after the earthquake, the average time spent on volunteer activities and hobbies tended to increase slightly, from before to after the earthquake (Table 4). This suggests that opportunities to participate in volunteer activities may have increased after the earthquake. However, the number of friends living nearby tended to decrease, suggesting that the disaster might have impacted the movement of residents.

**Table 4.** Responses to questions about participation in social activities.

| Questions | Before the Earthquake (Around 2010) | Five Years after the Earthquake (2016) |
|---|---|---|
| 1. The number of days in a month that you participate in community activities, such as neighborhood associations, children's associations, senior citizen associations, fire companies, etc. | 2.64 days | 2.47 days |
| 2. Average number of volunteer work hours per week (excluding community activities in 1.) | 0.785 h | 1.18 h |
| 3. Hours per week spent attending tea parties and other hobby meetings | 0.867 h | 1.00 h |
| 4. Hours spent talking to family members per day | 2.74 h | 2.77 h |
| 5. Hours spent talking to friends per day | 1.26 h | 1.10 h |
| 6. Average number of people greeted per day | 10.67 | 10.1 |
| 7. Number of friends living nearby | 8.13 | 6.41 |
| 8. Number of days per year spent attending events at neighborhood shrines, temples, and churches | 4.02 days | 4.35 days |
| 9. Number of new friends made at evacuation centers or places you moved to after the disaster | | 2.76 |

The above results show that after the earthquake, the SCs of local residents, mainly trust, were in a decline. This was because of the relocation of residents and temporary deterioration of security, and a decrease in the population engaged in agricultural work. By contrast, participation in community activities, such as volunteer work, increased, indicating that natural disasters can play a role in strengthening people's networks. This suggests that while traditional human relationships, such as land and blood ties, were severed, active network formation, such as participation in activities of one's own choosing, may have progressed.

*4.4. Regression Analysis of the Components of Trust*

Table 5 presents the regression analysis results. While most of the regression coefficients for explanatory variables related to household attributes, such as gender and participation in community farming, were not statistically significant, those with higher incomes and those who participated in hobbies, such as tea parties, tended to have higher trust indices. This result indicates that providing opportunities to participate in daily hobbies and tea gatherings can help foster trust. One reason that the coefficients of most of the individual attribute variables were not statistically significant is that the sample size was limited to less than 300 because some respondents did not answer every question.

Regarding characteristics related to disasters, moving to a different place and being a victim of crime immediately after an earthquake tended to lower trust in others. In a large-scale natural disaster that causes many residents to relocate and petty crimes to occur, maintaining local security and rebuilding resident networks after relocation is crucial.

**Table 5.** Results of regression analysis with trust index as the dependent variable (*n* = 288).

| Variables | Coefficient | *p*-Value |
|---|---|---|
| Gender | 0.021 | 0.886 |
| Farm | 0.042 | 0.766 |
| Community farm | −0.225 | 0.61 |
| Single | −0.275 | 0.234 |
| Age | −0.032 | 0.547 |
| Income | 0.126 *** | 0.000 |
| Move | −0.467 *** | 0.001 |
| Evacuate | −0.042 | 0.754 |
| Volunteer | 0.011 | 0.945 |
| Hobby | 0.385 ** | 0.013 |
| Damage | −0.009 | 0.815 |
| Crime | −0.285 * | 0.074 |
| Intercept | 1.523 *** | 0.000 |

R-squared adjusted for the degree of freedom: 0.144. f-value: 0.000. Statistical significance: *** significance level 1%, ** significance level 5%, * significance level 10%.

## 5. Conclusions

This study used a questionnaire survey of the coastal areas of Miyagi prefecture affected by the Great East Japan Earthquake to examine the changes in SC and local agriculture before and after the disaster and the factors behind these changes.

First, the results confirm that local agriculture was integrated into agricultural corporations or communal management as individual farmers left their farms. Consequently, the number of people participating in the region's cooperative management of farm roads and waterways has decreased. Second, regarding SC, the results show that trust in the surrounding environment was damaged by relocation after the disaster and exposure to crime. Furthermore, the results of the regression analysis suggest that to maintain or accumulate SC, which is also linked to the mental health of residents in a disaster area, it is important to maintain security in the disaster area and provide secure sources of income and opportunities for people to interact, such as hobby groups.

The findings of this large-scale survey study clarify the transformation of relational SC and local agricultural management in the recovery process following a large-scale natural disaster in a developed country with a declining population. This novel research subject has not been addressed by previous studies. In terms of efficient resource management, the results of this study support the "Blessings in Disguise" theory, which implies that disaster can encourage regional development, as proposed by Bănică et al. [43], in the sense that small farmers left the area and were consolidated into efficient, large-scale agricultural operations. However, from a microscopic viewpoint, the results also indicate that human relationships changed, mainly owing to the individual relocation of residents. Furthermore, social relational capital, which plays an important role in reconstruction and disaster prevention, was shown to have declined. The results of this study indicate that micro care is needed to promote the rebuilding of local communities to mitigate the negative effects of a large-scale natural disaster that involves the relocation of residents. The conditions affecting the cognitive SC of trust identified in this study can be applied to the development of specific care plans. Additionally, the findings can be used to provide administrative support in the recovery process after large-scale natural disasters that involve the relocation of populations, which is a frequent occurrence around the world today.

The limitations of this study are as follows. First, there were issues related to sample collection and bias. Although it was possible to distribute and collect the questionnaires in a highly anonymous manner, the collection rate was not high (about 10%), and there were many missing responses to questions, making statistical analysis difficult. Therefore, to conduct a more detailed statistical analysis, collecting accurate data through government agencies or agricultural organizations is necessary. Second, it was not possible to find a specific statistical relationship between regional agriculture and SCs. This can be resolved

by improving the previously mentioned sample collection methods and the bias and size of the sample population. In the future, it will be necessary to analyze the relationship between SCs and the management of local agricultural resources through the incorporation and scale expansion of agriculture. Furthermore, now that 12 years have passed since the earthquake, additional analysis, reflecting the long-term transformation of local agriculture and local communities, is desirable.

**Funding:** This research was funded by the Japanese Ministry of Education, Culture, Sports, Science and Technology Grants-in-Aid for Scientific Research (KAKENHI), Grant Numbers 26881002 and 23K12330, and Fuji Film Green Fund.

**Institutional Review Board Statement:** Ethical review not applicable.

**Informed Consent Statement:** Not applicable.

**Data Availability Statement:** All related data and methods are presented in this paper. Additional inquiries should be addressed to the corresponding author.

**Acknowledgments:** I am grateful to JA Sendai, the interviewees, and the respondents for their cooperation with the survey.

**Conflicts of Interest:** The author declares no conflict of interest.

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
