# Peer review of "Regional Agriculture and Social Capital after Massive Natural Disasters: The Case of Miyagi Prefecture after the Great East Japan Earthquake"

_sustainability, doi:10.3390/su151511725_

Round 1
Reviewer 1 Report
Dear Author,
Suggested corrections are;
“ I conducted a questionnaire survey in the Miyagi prefecture’s disas- 12 ter-affected areas “. Please don’t use I. You can use “In this study” or “the study”. Please correct this in all of the article.
The numerical results obtained should be added in the abstract with 1-2 sentences.
The importance of the study should be revealed by adding the importance of the effects of earthquakes and natural disasters at the beginning of the introduction, by giving some references. Some suggested references;
Cornell, C.A.: Engineering seismic risk analysis, Bulletin of the Seismological Society of America, 58 (1968) 5, pp. 1583-1606.
Coburn, A., Spence, R.: Earthquake Protection. Second Edition. John Wiley & Sons, Chichester, 2002,
Işık, E., et al (2019). Determination of Urban Earthquake Risk for Kırşehir, Turkey. Earth Sciences Research Journal, 23(3), 237-247.
Hoyois P., Below R., Scheuren J.M., Guha-Sapir D., (2007), Annual disaster statistical review: numbers and trends, Centre for Research on the Epidemiology of Disasters (CRED), School of Public Health, Catholic University of Louvain, Brussels, Belgium
IÅŸik, E., Kutanis, M., & Bal, I. E. (2017). Loss Estimation and seismic risk assessment in Eastern Turkey [Estimated loss and rating of earthquake risk in eastern Turkey].
The literature part is very important in order to reveal the novelty and diversity of the study. The literature section for such a study remained very weak. Please expand the literature section a little more in this context.
The novelty/difference of the study is clearly stated at the end of the introduction.
The reasons for the selected parameters should be added to the article.
You used we in some part of article. But there is only one author. Please check this.
Some figures have very low resolution. Please increase.
Limitations on the study should be stated.
The applicability of the study and how its usability will be in the future should be added to the conclusion part of the article.
Expand conclusion part.
Author Response
Thank you very much for your insightful comments. Please see the attachment for the response.

Reviewer 2 Report
The article entitled:
Regional Agriculture and Social Capital after Massive Natural Disasters: The Case of Miyagi Prefecture after the Great East Japan Earthquake presents an interesting pertinent analysis of a topical issue in its regional context.
The logic of the scientific approach and the motivation of the topic is well explained in its geographical context.
The methodology is convincing and appropriate for the chosen topic and results are concisely and clearly presented.
Even if I did not find major issues I recommend the author to take into consideration my below suggestions for an improved version of this paper in view of publication.
11. Please revise and extend literature review on the topic. Enlarge this chapter on both the topic of regional agriculture and the concept of social capital and their importance after massive natural disasters. Please add a consistent paragraph about the wide debated scientific term of resilience / community resilience. This complex notion could also be discussed with reference to study cases in different parts of the world which emphasize the communities’ regional agriculture and social capital post-disaster. In the light of this suggestion please take into consideration that a list of 13 titles for Bibliographic references is too short for a scientific paper in such a journal. A consistent bibliographic list should contain at least 25 – 30 titles.
22. Please considerably extend the exploitation on results and comments on the main findings and implications of the study. Chapter 4 is well structured but as entitled “results and discussion” it should contain more debates on results and the problematization of results in their context
33. Please revise writing in some figures (e.g., In Figure “20 –“ should be probably written “≥ 20”; on the pie charts – Figures 4 and 6 the writing is not uniform, appearing either on the graph area or outside the graph area)
Minor corrections are needed. Please provide a final proofreading by a native speaker.
Author Response

(The authors gave the same response as above.)

Round 2
Reviewer 1 Report
Dear Authors,
You have been made many corrections that suggested .
Yours Sincerely